# Frailty Syndrome in Older Adults with Cardiovascular Diseases–What Do We Know and What Requires Further Research?

**DOI:** 10.3390/ijerph19042234

**Published:** 2022-02-16

**Authors:** Marta Wleklik, Quin Denfeld, Magdalena Lisiak, Michał Czapla, Marta Kałużna-Oleksy, Izabella Uchmanowicz

**Affiliations:** 1Department of Nursing and Obstetrics, Faculty of Health Sciences, Wroclaw Medical University, 51-618 Wroclaw, Poland; marta.wleklik@umw.edu.pl (M.W.); magdalena.lisiak@umw.edu.pl (M.L.); izabella.uchmanowicz@umw.edu.pl (I.U.); 2Institute of Heart Diseases, University Hospital, 50-566 Wroclaw, Poland; 3School of Nursing, Oregon Health and Science University, Portland, OR 97239, USA; denfeldq@ohsu.edu; 4Laboratory for Experimental Medicine and Innovative Technologies, Department of Emergency Medical Service, Wroclaw Medical University, 51-616 Wroclaw, Poland; 51st Department of Cardiology, University of Medical Sciences in Poznan, 61-848 Poznan, Poland; marta.kaluzna@wp.pl

**Keywords:** acute coronary syndrome, cardiovascular disease, cardiac surgery, older adults, frailty syndrome, heart failure

## Abstract

Cardiovascular diseases (CVD) affect 60% of people over 60 years of age and are one of the main causes of death in the world. Diagnosed cardiovascular disease also triples the likelihood of Frailty syndrome (FS). FS has become increasingly relevant in cardiology and cardiac surgery and occurs in a significant number of patients with CVD, with prevalence ranging from 25% to 62%. Viewed in a multidimensional, biopsychosocial perspective, FS increases a patient’s vulnerability, making them susceptible to several adverse clinical outcomes. Frailty syndrome also is a predictor of mortality in patients with CVD regardless of age, severity of disease, multi-morbidity, and disability. Frailty syndrome potentially can be prevented in patients with CVD and its early identification is important to avoid the development of disability, dependence on others and reduced quality of life. The aim of this paper is to show the relationship between FS and specific CVDs (coronary artery disease, hypertension, atrial fibrillation, heart failure) and cardiac procedures (device implantation, cardiac surgery, and transcatheter aortic valve implantation). Furthermore, we highlight those areas that require further research to fully understand the relationship between FS and CVD and to be able to minimize or prevent its adverse effects.

## 1. Introduction

Cardiovascular diseases (CVD) affect a majority of people over 60 years of age and are one of the main causes of death in the world. It has been shown that CVD also triples the likelihood of Frailty syndrome (FS) [1]. Thus, as the older adult population grows, FS will be a significant social and medical problem [2]. Frailty syndrome is a geriatric syndrome characterized by age-related decrease of reserve capacity of various systems and lack of resilience to stressors [3]. It occurs in 25% to 62% of patients with CVD, with prevalence rates varying depending on the disease analysed and the FS definition applied [4,5]. Patients predisposed to developing FS have more CVD than non-frail patients [6]. Moreover, the presence of FS increases the risk for faster onset of any type of CVD (regardless of classical cardiovascular risk factors) and adds about 4-fold the risk of death from cardiovascular causes [7].

Studies indicate a two-way relationship between CVD and FS [8]. The pathophysiology of FS may be associated with metabolic imbalance and impaired functioning of the immune and endocrine systems [9], resulting in an imbalance between anabolic and catabolic states. Frailty syndrome is thus often associated with metabolic deficiencies, increased nutritional risk, and sarcopenia [10,11]. One hypothesized common pathophysiological pathway underlying both FS and CVD is a chronic inflammatory response [12]. The presence of chronic diseases, such as CVD, contributes to the stimulation of the immune system and sympathetic nervous system, inducing inflammation as manifested by elevated C Reactive Protein, white blood cell count, and interleukin-6 [13]. In addition, patients with FS have increased tumour necrosis factor α, fibrinogen, D-dimers, and tryptophan C-glycosyl levels; decreased vitamin D, sex hormones and growth hormone levels; and abnormal cortisol secretion [13,14]. Moreover, CVD with co-existing FS may develop gradually over several years and exhibit a long subclinical phase. For example, undiagnosed subclinical cardiovascular disorders such as myocardial injury, ischemic brain lesions, abnormal ankle-arm ratio, carotid artery stenosis, arterial hypertension, and left ventricular hypertrophy are more common in frail patients [15]. Finally, FS, regardless of the patient’s age, multi-morbidity status or medications taken, is related to the presence of cardiovascular risk factors [16]. Therefore, aggressive management of these factors may be an important element of breaking this vicious circle [17]. 

Frailty syndrome in patients with CVD potentially can be prevented and the early identification is important to avoid the development of disability, dependence on others and reduced quality of life [18,19]. For example, the pre-frail condition is associated with a four-fold higher risk of FS development in four years. This suggests that focusing on the pre-frail condition as a potentially reversible cardiovascular risk factor in older adults may have significant implications. Among the physical domains of pre-frailty, low gait speed seems to be the best predictor of future CVD [16]. Table 1 summarizes methods for assessing FS in various groups of patients with CVD. 

Given the significant intersection of FS and CVD among older adults, this paper summarizes the state of the science regarding the relationship between FS and specific CVDs (coronary artery disease, hypertension, atrial fibrillation, heart failure) and cardiac procedures (device implantation, cardiac surgery, and transcatheter aortic valve implantation (TAVI). Furthermore, we identify areas requiring further research to fully understand the relationship between FS and CVD and to be able to minimize or prevent its adverse effects (Table 2).

The scientific evidence underlying this publication was obtained from an analysis of papers indexed in the PubMed database. The search was limited to articles published between 2000 and November 2021. The search was limited to full-text papers published in English. The database was searched for relevant MeSH phrases and their combinations and keywords including: *“elderly, frail”; “frailty, elderly”; “frail older adults”; “frailty, older adults”; adult, frail older”; “frailty, heart failure”; “frailty, hypertension”; “frailty, atrial fibrillation”; “frailty, cardiac surgery”; “frailty myocardial infraction”; “frailty, acute coronary syndrome”; “frailty, coronary artery disease”; “frailty, implantable devices”; “frail, cardiovascular disease”; frailty, cardiovascular disease”; “frailty, TAVI”; “aging, cardiovascular diseases”; “older adults, cardiovascular diseases; “older adult, cardiovascular diseases”.*

## 2. FS and Coronary Artery Disease

There is a strong intersection of FS and coronary artery disease (CAD) [20,21,57]. For example, the Women’s Health Initiative Study showed that women with CAD are more likely to develop de novo FS within six years than women without CAD [57]. Lyu et al. reported that the prevalence of FS in patients with chronic coronary syndrome was about 30%, and age, malnutrition, hearing dysfunction, depression, and venous thromboembolism risk were significantly associated with FS [20]. Moreover, about half of patients with non-ST-elevation myocardial infarction have FS, which is associated with an increased risk of recurrent myocardial infarction, revascularization, hospitalization, bleeding, stroke, dialysis, and mortality, compared to patients without FS [22]. Frailty syndrome also leads to an increased incidence of cardiovascular events in patients after ST-elevation myocardial infarction, such as ischemic stroke or myocardial infarction. Singh et al. reported that among older adults who underwent percutaneous coronary intervention (PCI), 66% had some degree of FS and that the three-year mortality in frail patients was 28% compared to 6% in non-frail patients [58]. Garcia-Bias et al. showed that FS is independently associated with fewer days of out-of-hospital survival after acute myocardial infarction and higher rates of death and reinfarction [23]. A meta-analysis of 10 cohort studies involving 7,449,001 patients suggests that FS may be an independent risk factor for poor prognosis in patients with CAD after PCI. Specifically, FS was independently associated with higher rates of all-cause mortality and major adverse cardiovascular events, i.e., death, myocardial infarction, percutaneous coronary intervention, coronary artery bypass grafting, stroke, or transient ischemic attack [54]. Finally, in patients with CAD, FS is associated with poor quality of life, more comorbidities, and worse burden of cardiovascular symptoms; thus, FS should be considered alongside treatment of CAD to improve quality of life [21]. 

There are additional implications for treatment of CAD and acute coronary syndrome in the setting of FS among older adults. For example, standard 12-month dual antiplatelet therapy (DAPT) is usually recommended for patients undergoing PCI after ACS. However, the use of antiaggregants in frail older patients remains a challenge as FS is associated with in-hospital bleeding and predicts major bleeding within 30 days after hospital discharge, increasing all-cause mortality. Because of the increased risk of bleeding in frail patients undergoing PCI after ACS, a short DAPT is suggested. The PRECISE-DAPT (PREdicting Bleeding Complications In patients undergoing Stent Implantation and subsEquent Dual Anti Platelet Therapy) scale for stratifying bleeding risk at one year after PCI should be routinely evaluated as it serves to adjust DAPT recommendations [2]. Advanced age and FS are two common conditions for which the benefit-risk ratio of extending DAPT beyond 12 months must be carefully weighed in light of risk for ischemic events and hemorrhagic events. Thus, it is essential to perform an individualized assessment to determine the correct prognostic and therapeutic strategy [59]. 

## 3. FS in Hypertension

Hypertension is associated with an increased risk of FS [24]. In a systematic review and meta-analysis, it was estimated that about 14% of patients with hypertension were considered frail [25]. In longitudinal studies, the incidence of FS ranged from 3% to 16%, in cross-sectional studies, the incidence of FS ranged from 3% to 68% [25,60]. One consideration of FS among those with hypertension is that FS is associated with limited life expectancy. As described in the SHARE (The Survey of Health, Ageing and Retirement in Europe) life expectancy of people with FS at the age of 70 years ranges from 0.1 to 1.8 years for men and 0.4 to 5.5 years for women [61]. The data from this study are important in the context of hypertension treatment because in people with FS, the time to benefit from a given treatment may exceed life expectancy, which may undoubtedly alter the risk-benefit ratio [61]. However, as hypertension control may have an impact on FS trajectory, further studies are also needed [24]. 

In older adults, FS can affect the relationship between blood pressure (BP) and health outcomes. Gijon-Conde et al. assessed the relationship between FS, disability, and ambulatory BP among community-dwelling older adults. In this sample, 6% were frail and 8.1% were disabled. Individuals with FS had 3.5 mmHg lower daytime systolic BP, 3.3% less systolic BP dipping (nocturnal systolic BP decline), and 3.6 mmHg higher night time systolic BP compared with those without FS. These findings may help explain the higher mortality associated with low clinical systolic BP in weak older patients observed in epidemiological studies [62]. A cohort study by van Hateren et al. showed that participants with high BP and FS had lower mortality rates compared to frail patients with low BP [63]. In the study by Anker et al., FS was found in 61% of patients. Mean systolic and diastolic BP were the lowest in frail patients. These differences in BP by FS status were consistently observed in all gender and age categories of participants. After correction for socio-economic features, CVD and BMI risk factors, BP remained lower among pre-frail and frail participants compared with non-frail [27]. Thus, relatively low BP levels may not be beneficial or even harmful to frail older adults. In addition to the association with low BP, FS has been shown to be associated with a higher risk of orthostatic hypertension. According to the guidelines, BP thresholds and treatment goals for BP should consider FS when deciding how to treat a patient [27].

Considering the physiological changes resulting from ageing and the presence of many coexisting diseases, the treatment of hypertension in older adults poses a major challenge for therapeutic teams [64]. In clinical practice, special attention should be paid to frail older patients, who should receive tailored treatment. The 2017 Hypertension Clinical Management Guidelines indicate that BP lowering therapy is one of the few interventions that contribute to reducing the risk of mortality in older adults with FS, but without providing specific recommendations for treating hypertension in this group. The Systolic Blood Pressure Intervention Trial (SPRINT) showed that intensive systolic BP control (less than 120 mmHg) leads to a reduction in cardiovascular events in people with both FS and non-fs compared to the group of patients with the standard systolic BP treatment goal (less than 140 mmHg). Frailty syndrome in the SPRINT study was assessed by FS index and gait speed test [26]. Interestingly, results from the SPRINT study showed that while targeting systolic BP below 120 mmHg reduces the incidence of cardiovascular events, it has no effect on gait speed (95% CI, −0.005 to 0.005; *p* = 0.88) and was not associated with change in mobility limitation (hazard ratio, 1.06; 95% CI, 0.92–1.22) [65]. The results from the SPRINT trial provide some clinical implications regarding the future of intensive BP therapy, especially among older adults who are at high absolute risk for cardiovascular complications associated with high BP values and the long-term consequences of these events [26]. There is little direct evidence of the risks and benefits of antihypertensive drugs when FS occurs. For frail older patients, treatment should be considered if systolic blood pressure (SBP) is 160 mmHg or more. If the patient is very frail and has a short life expectancy, an SBP target of 160–190 mmHg may be justified. If the SBP is below 140 mmHg, antihypertensive medication may be reduced, unless indicated in other conditions. Generally, no more than two antihypertensive drugs should be prescribed to avoid unnecessary administration of large numbers of drugs [66]. 

When implementing therapy among older adults, it is important to keep in mind that, due in part to the comorbidity of FS, this population is particularly vulnerable to the adverse effects of polypharmacy. The issue of polypharmacy is becoming increasingly important with respect to older patients, particularly during hospitalization because this is the period when medication errors and potential adverse effects can be identified. Unlu et al. showed that half of older hospitalized patients with heart failure (HF) take at least 10 medications (the cut-point for polypharmacy) at admission and at discharge, and each additional comorbidity increases the relative risk of polypharmacy at hospital discharge by 13% [67]. The significant increase in polypharmacy over time reflects the urgent need to develop appropriate strategies to mitigate the negative effects of this phenomenon and secondly to optimize the prescribing of noncardiac medications in the older patient population. The care of patients with multimorbidity and polypharmacy should be a formal process based on problem identification, communication, and provision of care that is evidence-based and consistent with patients’ health priorities [67]. 

Interestingly, the coexistence of FS may have a negative impact on compliance in older patients with coexisting hypertension [28]. Coexistence of FS and/or cognitive dysfunction among older adults is associated with a poorer perception of health, a comorbidity which may hinder compliance with therapeutic recommendations. In the Koizumi et al. study, FS in patients with hypertension was associated with reduced physical activity, lower body weight, difficulties in performing daily and complex daily activities, which translated into the treatment and control of hypertension [29]. Other data show that patients who did not have FS were more compliant with the Mediterranean diet than patients with FS [66]. In another study, Uchmanowicz et al. showed that FS was diagnosed in 66% of patients with hypertension, and the most important contributing factor was the social component of FS. Frailty syndrome was correlated with all subcomponents of the questionnaire, which evaluated the adherence to therapeutic recommendations in hypertension. Frailty syndrome interacted with the social FS subscale of the questionnaire, which concerned “reduced sodium intake”, “scheduled meeting”, and “taking antihypertensive drugs” [28].

## 4. Frailty Syndrome and Atrial Fibrillation

Atrial fibrillation (AF) is the most common arrhythmia, which is associated with a 5-fold increase in the risk of stroke, and its prevalence increases with age, affecting up to 4.2% of those aged 60–70 years and 17% of those aged 80 years or older [30]. Atrial fibrillation also is associated with co-existence and development of FS [68]. The prevalence of FS in patients with AF ranged from about 6% in the outpatient registry to about 100% in the nursing home population [30]. A systematic review showed that the prevalence of FS in patients with AF is as high as 75% [56]. The disparity in the prevalence of FS in AF is most likely dictated by the use of different tools to assess FS across studies [34,56]. In hospitalized older adults, AF is significantly associated with FS, which increases the risk of stroke incidence, falls, mortality, symptom severity and length of hospital stay [35]. Additionally, studies have demonstrated that patients with AF who were frail had a higher mean CHA2DS2-VASc and HAS-BLED scores than those who were non-frail [31,34,35,69].

Studies on the mechanisms linking FS and CHA2DS2-VASc score in patients with AF are limited [36,70]. The PURE-Rhythm score update noted that the CHA2DS2-VASc score expresses complex clinical status and is inversely correlated with the Short Physical Performance Battery and Frailty Index. Furthermore, a higher CHA2DS2-VASc score correlates with higher levels of inflammatory cytokines. The study authors suggested that in AF, inflammation may mediate the relationship between CHA2DS2-VASc score and FS [32]. Atrial fibrillation is also associated with reduced exercise capacity, heart rate, weakness, cognitive and vascular functions, all of which may contribute to the observed decrease in walking speed and FS [71]. 

Unfortunately, the optimal treatment strategy of patients with AF and coexisting FS remains unclear, especially given the heterogeneity in this patient population. The National Institute for Health and Care Excellence guidelines recommend the use of CHA2DS2-VASc to identify patients with a high risk of embolism [35]. Frailty syndrome, however, might influence the management decisions in older patients with AF, especially since CHA2DS2-VASc scores have not been validated in the context of FS [35]. In this group of patients with AF and FS, the implementation of oral anticoagulation deserves special attention given the fear of iatrogenic harm, falls, and FS in the older adult population. Oral anticoagulants are effective in preventing stroke in patients with AF, but they are not fully utilized in older adults, probably due to the misleading perception of FS. Patients with FS receive less anticoagulant therapy, especially when such common conditions as malnutrition, comorbidity, polypharmacy, cognitive dysfunction, depression occur [72,73] According to Zathar et al., FS is one of the most frequently quoted reasons for prescribing insufficient OAC, while patients with FS may actually benefit most from it [30].

This leads to a situation where the prevalence of FS in patients hospitalized with AF is high and the use of OAC remains low. Age alone cannot be a contraindication to the implementation of anticoagulant therapy because the absolute risk of intracranial haemorrhage in older patients taking anticoagulants is low, about 0.2% per year [72]. Studies have shown that patients with cognitive impairment and FS were less frequently prescribed OAC, while in fact they had a higher predicted stroke risk and observed mortality [74]. Further analyses have found no association between OAC use in the presence of cognitive dysfunction or FS and the incidence of mortality, major bleeding, stroke, non-central nervous system systemic embolism, TIA, myocardial infarction or cardiovascular death [74]. In fact, the presence of FS should not be the only determinant, and therefore setting the limit above which OAC would not be prescribed may be unreasonable, as it may lead to unjustified exclusion of treatment and constitute a barrier to individualised treatment [30]. In the ENGAGE AF-TIMI 48 double-blinded double-dummy trial, two once-daily regimens of edoxaban (a DOAC) were compared with warfarin. Edoxaban was similarly efficacious to warfarin across the FS spectrum and was associated with lower rates of bleeding except in those with severe FS [55]. 

## 5. FS and Heart Failure

According to a systematic review and meta-analysis by Denfeld et al. the prevalence of FS in patients with heart failure (HF) has been estimated to affect about 45% of the population, and the prevalence of FS was not dependent on age and functional classification [33]. There is a two-way relationship between FS and HF. Patients with HF are up to six times more likely to have FS, and people with FS have a significantly increased risk of developing HF [37].

Moreover, FS contributes to increased mortality and hospitalization and reduced quality of life in HF [12]. Co-occurrence of FS with other comorbidities in HF is also a common occurrence. For example, anaemia is common in frail hospitalized older adults with HF and has a negative impact on mortality; as such, both anaemia and FS may be important therapeutic targets in these patients [75]. Finally, the pathophysiological mechanisms underlying FS in HF are largely hypothesized, but likely relate to increased inflammation, impaired neurohormonal function, and/or metabolic/skeletal muscle dysfunction [11,15].

There are multiple domains of FS that can be assessed among older patients with HF [38]. The FRAGILE-HF prospective multicentre cohort study described the prevalence, overlap, and prognostic implications of physical and social frailties and cognitive dysfunction in hospitalized older patients with HF [39]. Physical FS was reported in 56% of patients, social FS in 66.4% of patients, and cognitive FS in 37% of patients [39]. Social FS, in particular, adversely affects self-care in older patients with HF [42] and is linked with poor outcomes in HF [39]. Thus, multidimensional assessment of FS in patients with HF may be helpful given the association of multiple FS domains with both adverse treatment outcomes and reduced access to and tolerance of therapy [37]. A multidimensional HF-specific tool was developed based on four main domains: clinical, psycho-cognitive, functional, and social [38]. 

It is believed that FS can be prevented or even reversed with appropriate interventions [63]. Taking a broad approach to FS encompassing the physical, psychological, and social realms, there are many interventions to prevent or reduce FS and prevent or delay its adverse effects. As indicated by the ESC guidelines for HF, treatment of FS in HF should be multifactorial and may include physical rehabilitation, nutritional supplementation, and an individualized approach to the treatment of comorbidities [37]. These strategies should focus also on polypharmacy and adherence improvement, falls prevention, mood, and cognitive interventions [76]. Physical activity is an important aspect of coping with FS. A review of randomized controlled trials confirmed the effects of physical training on improving muscle strength, body composition, mobility, functional status, and fall prevention [77].

Frailty syndrome often co-exists with advanced HF specifically, which is also accompanied by general muscle weakness, and later in the disease course, cardiac cachexia develops. In a study by Dunlay et al. of patients after left ventricular assist device implantation, patients with diagnosed FS had a 3-fold increased risk of death and an almost 1.5-fold higher risk of re-hospitalization, regardless of age or the Interagency Registry for Mechanically Assisted Circulatory Support profile [41]. In patients with diagnosed FS after heart transplantation, the survival rate after one year of observation was 52% ± 23% compared to 100% survival rate in non-frail patients. Moreover, patients with FS tended to stay longer in the intensive care unit (8 days vs. 6 days without FS) and presented a greater total length of hospitalization (27 vs. 24 days) [11,40]. A unique aspect of FS assessment in patients with advanced HF is that phenotype may be stimulated by HF itself and the resulting clinical and subclinical organ dysfunction, independent of age and associated pathophysiological pathways. This simulation encourages considerations of the extent to which FS coexisting with HF can potentially be reversible in LVAD patients. Furthermore, because FS includes vulnerability to stress and adverse outcomes, it may be an important factor in patient selection before LVAD implantation. 

## 6. FS and Implantable Devices

Implantable devices are effective methods of treating serious heart rhythm disorders. Approximately 45% of patients receiving cardiac resynchronization therapy (CRT) and 28% of patients receiving an implantable cardioverter-defibrillator (ICD) are 75 years of age or older. Moreover, at least 20% of patients have three or more chronic diseases, and the benefits of the ICD are inversely proportional to the number of concurrent diseases [78]. Hence, the increasing age and high number of comorbidities significantly increases the risk of FS among patients receiving an implantable device. However, the role of implantable devices in frail older adults has not yet been fully investigated [5]. Vetta et al. emphasized that older patients with comorbidity, cognitive dysfunction or FS were under-represented in clinical trials to assess the effectiveness of CRT and ICD [79]. 

Data on the prognostic effects of FS are usually derived from observational studies. In one retrospective study of patients undergoing primary prevention ICD implantation, frail patients had a 22% increased risk of death one year after implantation compared with non-frail patients, and the risk of death increased in cases of co-occurrence of dementia, diabetes, or chronic obstructive pulmonary disease [49]. In the Kramer et al. study, FS was identified in 12.8% of outpatients with an implantable device [43]. They also pointed out that the physical activity detected by the device may be clinically useful in identifying patients at risk of FS and adverse events. Patients with FS had a lower average walking speed (<0.8 m/s), and a significant relationship between FS and patient mobility was noted. An increase in daily activity of 1 h on average was associated with a 46% reduction of FS. The increase in activity detected by the device was associated with an increase in walking speed [43].

The results of the Mlynarska et al. study indicated that about 75% of older patients qualified for ICD implantation were affected by FS. In the FS subgroup, adequate and inadequate shocks occurred more often compared to the robust patients [44]. Notably, a systematic review by Chen et al. concluded that people with FS may not benefit from ICD implantation as part of primary prevention of sudden cardiac death, as higher annual mortality and increased dementia occurs after ICD implantation among patients with FS [44]. According to the European Society of Cardiology guidelines, ICD implantation is recommended as a primary prevention to reduce the risk of sudden death and all-cause mortality in patients with symptomatic HF of ischemic aetiology, provided they are expected to survive significantly longer than 1 year in good functional status [37]. As FS is often associated with functional decline, it is important to screen for FS in older patients who are considering ICD implantation as part of primary prevention [49]. As highlighted by the authors of the European Heart Rhythm Association study, traditional clinical survival predictors seem to be losing their relevance among older adults who received an ICD [80]. Identification of FS may help determine appropriate ICD use and contribute to improved patient outcomes. 

Identifying patients who may benefit from CRT is important because implantation of the CRT-D is still a procedure that is expensive, invasive, and carries a significant risk. In a study of older patients hospitalized for CRT implantation, FS was recorded in 75.64% of patients, using a multidimensional tool, Tilburg Frailty Indicator. Complications and electrical storm were reported in patients with identified FS, but none in patients without FS [47]. A study by Dominiguez-Rodriguez et al. evaluated the relationship between baseline FS and clinical outcomes 12 months after CRT-D implantation in patients with ischaemic cardiomyopathy. Before implantation, 28% of patients were frail, and FS was a strong predictor of adverse results after CRT-D implantation [45]. Moreover, they showed that FS was associated with a 4.5-fold increase in decompensated HF [45]. Finally, FS was shown to be associated with a reduced likelihood of response to CRT at 12 months. Specifically, frail patients were less likely to have at least one NYHA class reduction, hospitalization for heart failure, and an absolute improvement of 10% in baseline left ventricular ejection fraction compared with non-frail patients [48]. Taken together, FS should be evaluated alongside other clinical factors when considering CRT implantation.

According to the results of the European Heart Rhythm Association Survey, which assessed FS and its impact on the clinical management of arrhythmias in 41 European centres, FS or pre-frail was found in less than 10 percent of patients who received implantable devices, and pacemakers were the most used devices in patients with FS [80]. Pacemakers for bradyarrhythmia control should be strongly considered for patients with FS as a pacemaker can reduce the risk of falls, improve or maintain quality of life and prevent deterioration of HF by maintaining heart rhythm and subsequent deterioration of major organ function [81]. According to the decision algorithm in the ESC guidelines, a pacemaker should be implanted in patients with an unexplained syncope who are older, have FS and are at risk of traumatic recurrence, when they have bundle branch block, bifascicular block and whose left ventricular ejection fraction is above 35% [82]. 

## 7. FS and Cardiac Surgery and TAVI

Frailty syndrome is of particular importance in cardiac surgery because an operation itself is a stressor, and the perioperative course is largely dependent on the patient’s initial immunity and physiological reserves [83]. In older patients undergoing cardiac surgery, FS can be as high as 50% [17]. In the FS Assessment Before Cardiac Surgery study, 46% of patients aged 70 years or older undergoing coronary artery bypass and/or heart valve surgery were frail in a 5-m gait speed test [4]. Frailty syndrome is also associated with postoperative complications after cardiac surgery. Postoperative complications in patients after scheduled coronary artery bypass surgery were reported in 17% of patients without identified FS, 28% of patients with identified pre-frail condition and 56% of patients with FS regardless of chronological age [51]. A meta-analysis showed that FS and pre-frailty were associated with greater adjusted operative mortality and adjusted perioperative complications, and FS was associated with almost 5-fold risk of non-home discharge [49]. Preoperative FS also predicts prolonged respiratory therapy, prolonged stay in the intensive care unit, functional decline, postoperative delirium, reoperations, readmission rate, increased risk of institutional discharge, and early, mid-term, and late mortality [50,52,80,84,85,86,87,88,89,90,91,92,93,94].

Identification of FS has still not been integrated into daily clinical practice and included in commonly used risk assessment models. The addition of FS increases the predictive power of conventional perioperative risk assessment models in cardiac surgery. The 5-m gait speed test is recommended by the STS Society as a measure of physical FS in patients scheduled for cardiac surgery given the strong evidence of the predictive value of this test. For example, Afilalo et al. demonstrated that walking a distance of 5 m in more than 6 s is associated with an increased risk of cardiac surgery complications, including an 8-fold increase in morbidity and mortality in older women [11,53,95]. However, the gait speed test is a unidimensional tool [18]. There is still a lack of consensus that identifies a specific multidimensional tool for assessing FS in cardiac surgery. The predictive value of classical perioperative risk assessment models may be significantly improved when multidimensional FS assessment is included [12]. Afilalo et al. emphasized the value of FS and disability and the need to consider them in addition to classical risk assessment models to more accurately predict morbidity and mortality after cardiac surgery [53]. For patients with FS at high risk of adverse events or where the balance of risks and benefits may be ambiguous, the efficacy of many traditional cardiac surgery procedures becomes questionable. In this particular patient population with increased risk of morbidity and mortality, risk stratification is even more important, especially since cardiac surgery now has alternatives to conventional cardiac surgery and less invasive methods such as TAVI [96].

Transcatheter aortic valve implantation has become a less invasive alternative to surgical aortic valve replacement in patients with severe aortic valve stenosis at a high risk of perioperative surgery. In TAVI patients, the prevalence of FS as assessed by various tools is twice as high as in SAVR patients [94]. Frailty syndrome also is a strong predictor of outcomes after TAVI, especially given that these patients have high perioperative risk, advanced age, left ventricular dysfunction, and a number of comorbidities [97]. Green et al. found no influence of FS on perioperative complications, but FS was associated with a higher risk of 1-year mortality [98]. According to the EuroSCORE (The European System for Cardiac Operative Risk Evaluation) and STS (The Society of Thoracic Surgeons) models, FS may be more useful in predicting functional decline after TAVI than perioperative risk assessment [97]. 

Given that FS or functional incompetence is commonly diagnosed among older adult patients before traditional and minimally invasive cardiac surgical procedures, actually confirms the need to routinely take this condition into account in the eligibility procedure [53]. Early, preoperative preparation of the patient for elective cardiac surgery (cardiac prehabilitation) aims to increase the physiological reserve and resilience of the patient with coexisting FS to the onset of perioperative stress [16]. Appropriately planned cardiac rehabilitation programs may help increase functional capacity, exercise ability, psychosocial well-being, nutritional status, increase level of independence, and reduce risk of death [82]. There is ongoing research related to testing various preoperative strategies, including physical therapy, supplementation, and anxiety reduction, to better prepare patients and improve surgical outcomes [80]. Due to the multidimensional nature of FS, interventions that focus simultaneously on several components of FS have been attributed with greater efficacy. An ongoing randomized, multicentre clinical trial is investigating the feasibility and value of a multidisciplinary program that includes nutritional support, 8 weeks of physical training, and education in a group of older patients scheduled for elective cardiac surgery [99]. 

## 8. Conclusions

The concept of FS has become increasingly popular in cardiology and cardiac surgery. The prevalence of FS in patients with CVD and its proven impact on the occurrence of many adverse medical consequences should therefore be associated with the systematic identification of FS in clinical practice and the introduction of targeted interventions to optimize the status of patients with CVD and coexisting FS. Frailty syndrome potentially is a reversible condition, so there is a need to establish appropriate management strategies, as well as to adopt appropriate definitions and tools for identification of FS in cardiology and cardiac surgery.

## Figures and Tables

**Table 1 ijerph-19-02234-t001:** Tools for assessing Frailty syndrome in various groups of CVD, based on articles reviewed.

Coronary artery disease	Frail scale [20]; Frailty Phenotype [21]; Clinical Frailty Scale [22,23]; The Edmonton Frail Scale [21];
Hypertension	Frail scale [24]; Frailty Index [25,26]; Clinical Frailty Scale [25]; Frailty Phenotype [25,27]; Gait speed [26]; Tilburg Frailty Indicator [28]; The Basic Checklist for Frailty—BCF [29];
Atrial fibrillation	Frailty Phenotype [30]; Frailty Index [30,31,32,33]; Clinical Frailty Scale [34,35];The Edmonton Frail Scale [34,35]; Tilburg Frailty Indicator [35]; Frail scale [35]; Gait speed [26,36];
Heart Failure	Frailty Phenotype [37,38,39,40], Gait speed [33]; Frailty index [37,41]; Tilburg Frailty Indicator [37,42]; Aging Clinical Frailty Scale [33]; Handgrip strength [33];
Implantable devices	Gait speed [43]; Frailty Phenotype [43,44,45]; The Edmonton Frail Scale [21]; Tilburg Frailty Indicator [46,47,48];
Cardiac surgery and TAVI	Frailty Phenotype [49,50], Gait speed [4,51,52,53] The Essential Frailty Toolset [51]; Frailty index [49]; Clinical Frailty Scale [49]; Clinical Frailty Scale [49]; Simplified Comprehensive Assessment of Frailty [50]; Modified Fried Frailty Criteria [50]; Frailty scale [53];

**Table 2 ijerph-19-02234-t002:** Areas that require further research to fully understand the relationship between Frailty syndrome and cardiovascular disease.

Coronary artery disease	■ Determining the best and optimal methods to identify frailty to develop individualized models of care [2], as well as develop interventions to reduce or reverse frailty status [20,54]; ■ Evaluating whether strategies to reduce frailty may provide additional clinical benefit [54]; ■ Comprehensive evaluation of high-risk patients to identify determinants of frailty progression [20];
Hypertension	■ Understanding the relationship between BP and FS to enable providers to treat hypertension appropriately [27]. ■ Assess in future prospective clinical trials whether intensive control of hypertension can affect the frailty trajectory [24]; ■ Defining the target blood pressure ranges for patients with hypertension and coexisting frail.
Atrial fibrillation	■ Conduct high-quality, population-specific randomized controlled trials in patients with frailty and AF to guide therapy in this vulnerable population [55]; ■ Assessing the relationship between AF and FS and to determine the optimal therapeutic approach to AF in frail individuals [56]; ■ Evaluating FS as a trigger of HF and studies on the inclusion of FS in the CHA2DS2VASc score.
Heart Failure	■ Despite the knowledge that frailty is associated with poorer outcomes among patients, more research is needed on the mechanistic relationship between FS and HF; ■ Establishing future interventions that prioritize patient-centred outcomes with a focus on frailty may become an important strategy aimed at improving clinical prognosis and outcomes in the growing population of frail older adults with HF.
Implantable devices	■ Assessing whether frailty affects the risks and benefits of implanted devices;
Cardiac surgery and TAVI	■ Improve models that predict perioperative risk, and lead efforts to introduceroutine FS assessment into the qualification process; ■ Prehabilitation models and their effectiveness before cardiac surgery to improve outcomes with frailty;

## Data Availability

The data will be available by contacting the corresponding author.

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
