# Peer review of "Frailty Syndrome in Older Adults with Cardiovascular Diseases–What Do We Know and What Requires Further Research?"

_ijerph, 2022, doi:10.3390/ijerph19042234_

Round 1

Reviewer 1 Report

The importance of identifying and addressing Frailty syndromes in patients with various cardiovascular diseases especially when they are candidates for interventions is well established nowadays, but specific guidance is missing. In these terms the work of the authors is of great significance. 

The manuscript is quite extensive and several points are repeated more than once. The authors should try to organise the content and make it shorter perhaphs using tables to clarify the work previously reported. 

I would suggest to include a paragraph after Introduction with a Table and short description of methods used to assess Frailty in various groups of CVD patients. Furthermore, it is reported repeatedly that Frailty can be prevented but reference to the prevention of Frailty is very limited; please elaborate.

Author Response

REVIEWER 1

Comments and Suggestions for Authors

Reply

The importance of identifying and addressing Frailty syndromes in patients with various cardiovascular diseases especially when they are candidates for interventions is well established nowadays, but specific guidance is missing. In these terms the work of the authors is of great significance.

The authors are grateful for the reviewer's emphasis on the importance of the work topic undertaken in this review.

The manuscript is quite extensive and several points are repeated more than once. The authors should try to organise the content and make it shorter perhaphs using tables to clarify the work previously reported.

In accordance with the reviewer's suggestion, it was decided to put the repeated section "Future Research" into one, collective table. The contents of the table were arranged in such a way that they would not be repeated.

I would suggest to include a paragraph after Introduction with a Table and short description of methods used to assess Frailty in various groups of CVD patients. Furthermore, it is reported repeatedly that Frailty can be prevented but reference to the prevention of Frailty is very limited; please elaborate.

As suggested by the reviewer, a table presenting methods for assessing frailty syndrome in different CVDs was included after the "introduction" section. In addition, the content on the prevention of FS has been completed as suggested. These are located in sections: "FS in Heart Failure" and "FS in cardiac surgery and TAVI".

Reviewer 2 Report

The presented article focused on „Frailty syndrome in older adults with cardiovascular diseases. What do we know and what requires further research?“. There is a significant amount of research on this subject and it is discussed extremely frequently, demonstrating the importance of this topic. The aim of the review was to demonstrate the relationship between frailty syndrome and different aspects like coronary artery disease, hypertension, atrial fibrillation, heart failure, implantation of implantable devices, cardiac surgery procedures, and TAVI. All aspects were presented independently. The authors concluded: The prevalence of FS in cardiovascular diseases and its proven impact on the occurrence of many adverse medical consequences should therefore be associated with the systematic identification of FS in clinical practice and the introduction of targeted interventions to optimize the status of patients with cardiovascular disease and the coexisting FS. Frailty syndrome is a reversible condition, so there is a need to establish appropriate management strategies, as well as to adopt appropriate definitions and tools for identification of FS in cardiology and cardiac surgery.

However, there are some drawbacks in the manuscript that hinder its immediate publication.

But in detail:

Language: in total good, but some shortcomings in type writing were found: e.g. often too many blanks between the words.

The manuscript is very detailed in several parts but often redundant. Therefore, many aspects are brought up several times, which leads to a very long text. This takes the excitement out of the article and bears the risk that the reader loses the information content. Since many papers come to the same conclusion or deal with the topic almost identically, a table can be useful for streamlining (e.g. compare the editorial of von Haehling et al (2013). Frailty and heart disease. doi:10.1016/j.ijcard.2013.07.068). A "red thread" would be important and streamline the article.  Overall, many of the cited papers are superficially presented. For example, p-values, CI, number of patients, or whether the studies are randomized trials are missing. This does not have to be provided for all articles, but for the most important ones. Furthermore, the search mechanism is missing how the presented articles were selected or discarded.  In which database was searched? A flow chart could illustrate this clearly.  

On the other hand, it would be interesting to discuss the phatophysiological causes of frailty in more detail with 2 or 3 sentences rather than referring solely to inflammation.

It seems that some references are not listed, e.g. in line 203 "Koizumi et al." or in line 215 "van Hateren et al". The following papers are probably intended: Koizumi Y, Hamazaki Y, Okuro M, et al. Association between hypertension status and the screening test for frailty in elderly community-dwelling Japanese. Hypertens Res. 2013;36(7):639-644. doi:10.1038/hr.2013.7 and Van Hateren KJ, Hendriks SH, Groenier KH, et al. Frailty and the relationship between blood pressure and mortality in elderly patients with type 2 diabetes (Zwolle Outpatient Diabetes project Integrating Available Care-34). J Hypertens. 2015;33(6):1162-1166. doi:10.1097/HJH.0000000000000555

Author Response

Reviewer 2

Comments and Suggestions for Authors

Reply

The presented article focused on „Frailty syndrome in older adults with cardiovascular diseases. What do we know and what requires further research?“. There is a significant amount of research on this subject and it is discussed extremely frequently, demonstrating the importance of this topic. The aim of the review was to demonstrate the relationship between frailty syndrome and different aspects like coronary artery disease, hypertension, atrial fibrillation, heart failure, implantation of implantable devices, cardiac surgery procedures, and TAVI. All aspects were presented independently. The authors concluded: The prevalence of FS in cardiovascular diseases and its proven impact on the occurrence of many adverse medical consequences should therefore be associated with the systematic identification of FS in clinical practice and the introduction of targeted interventions to optimize the status of patients with cardiovascular disease and the coexisting FS. Frailty syndrome is a reversible condition, so there is a need to establish appropriate management strategies, as well as to adopt appropriate definitions and tools for identification of FS in cardiology and cardiac surgery.

The authors are grateful for the reviewer's emphasis on the importance of the work topic undertaken in this review.

Language: in total good, but some shortcomings in type writing were found: e.g. often too many blanks between the words.

In regards to the reviewer's suggestion, shortcomings in the writing have been corrected.
Too many blanks between the words have been corrected and eliminated.

The manuscript is very detailed in several parts but often redundant. Therefore, many aspects are brought up several times, which leads to a very long text. This takes the excitement out of the article and bears the risk that the reader loses the information content. Since many papers come to the same conclusion or deal with the topic almost identically, a table can be useful for streamlining (e.g. compare the editorial of von Haehling et al (2013). Frailty and heart disease. doi:10.1016/j.ijcard.2013.07.068). A "red thread" would be important and streamline the article.  Overall, many of the cited papers are superficially presented. For example, p-values, CI, number of patients, or whether the studies are randomized trials are missing. This does not have to be provided for all articles, but for the most important ones. Furthermore, the search mechanism is missing how the presented articles were selected or discarded.  In which database was searched? A flow chart could illustrate this clearly.  

In accordance with the reviewer's suggestion, it was decided to put the repeated section "Future Research" into one, collective table. The contents of the table were arranged in such a way that they would not be repeated. With a table, the answer to the question posed in the title of the manuscript becomes more exposed. Furthermore, a table has also been added that presents tools for assessing FS in various CVD.

As suggested by the reviewer, some of the cited papers were supplemented with information on the type of study and p-values, HR and CI were added.

As suggested by the reviewer, the "Introduction" section includes content on how the papers included in this review were searched.

On the other hand, it would be interesting to discuss the phatophysiological causes of frailty in more detail with 2 or 3 sentences rather than referring solely to inflammation.

As suggested by the reviewer, the content relating to the pathophysiological causes of frailty was expanded.

It seems that some references are not listed, e.g. in line 203 "Koizumi et al." or in line 215 "van Hateren et al". The following papers are probably intended: Koizumi Y, Hamazaki Y, Okuro M, et al. Association between hypertension status and the screening test for frailty in elderly community-dwelling Japanese.
Hypertens Res. 2013;36(7):63 644.doi:10.1038/hr.2013.7 and Van Hateren KJ, Hendriks SH, Groenier KH, et al. Frailty and the relationship between blood pressure and mortality in elderly patients with type 2 diabetes (Zwolle Outpatient Diabetes project Integrating Available Care-34). J Hypertens. 2015;33(6):1162-1166.doi:10.1097/HJH.0000000000000555

As noted by the reviewer, two articles were not included in the references. As suggested by the reviewer, both items were cited and saved in the references at positions 39 and 40.

Round 2

Reviewer 2 Report

Dear Ladies and Gentlemen,

First of all, many thanks to the authors for editing the manuscript. Unfortunately, some points are open. As already described in the first review, many passages are redundant. Therefore, the manuscript became very long and made it difficult to read. Because of the redundant phrases, the review loses some of its punch. In addition, it is somewhat hard for the reader to stay focused on the manuscript, because many passages contain common statements, some of which are just worded differently.

For example: It is clear that therapies for older patients are difficult and they should be treated with caution. Line 144 "..frail elderly patients...., a short DAPT is suggested." Lines 148-150: "Advanced age and frailty....extending DAPT beyond 12 months must be carefully weighted. Lines 150-153: "...DAPT.... may cause in elderly....affected by frailty must be carefully considered. Lines 153-154: "Proper individualized assessment...elderly....essential to determine...."

As stated, this is just one example of many. This style of writing runs throughout the manuscript. The table is actually quite good for that purpose, but also in the text the merging of statements from several authors and from generalized statements should be continued.

Please read the first review: “Since many papers come to the same conclusion or deal with the topic almost identically, a table can be useful for streamlining (e.g. compare the editorial of von Haehling et al (2013). Frailty and heart disease. doi:10.1016/j.ijcard.2013.07.068).

Therefore, the manuscript should be revised.

Author Response

Thank you very much for this remmark. We have carefully and thoroughly edited the manuscript to reduce the redundant language and to convey our message more clearly.
